# Functional modulation of primary visual cortex by the superior colliculus in the mouse

Mehran Ahmadlou [1], Larry S. Zweifel[2,3] & J. Alexander Heimel [1]

The largest targets of retinal input in mammals are the dorsal lateral geniculate nucleus (dLGN), a relay to the primary visual cortex (V1), and the superior colliculus. V1 innervates and influences the superior colliculus. Here, we find that, in turn, superior colliculus modulates responses in mouse V1. Optogenetically inhibiting the superior colliculus reduces responses in V1 to optimally sized stimuli. Superior colliculus could influence V1 via its strong projection to the lateral posterior nucleus (LP/Pulvinar) or its weaker projection to the dLGN. Inhibiting superior colliculus strongly reduces activity in LP. Pharmacologically silencing LP itself, however, does not remove collicular modulation of V1. The modulation is instead due to a collicular gain modulation of the dLGN. Surround suppression operating in V1 explains the different effects for differently sized stimuli. Computations of visual saliency in the superior colliculus can thus influence tuning in the visual cortex via a tectogeniculate pathway.

[1] Department of Cortical Structure & Function, The Netherlands Institute for Neuroscience, Royal Netherlands Academy of Arts and Sciences (KNAW), Meibergdreef 47, 1105 BA Amsterdam, The Netherlands. [2] Department of Pharmacology, University of Washington, Seattle, WA 98195, USA. [3] Department of Psychiatry and Behavioral Science, University of Washington, Seattle, WA 98195, USA. Correspondence and requests for materials should be addressed to J.A.H. (email: heimel@nin.knaw.nl)

In mammals, visual information flows from the retina to the dorsal lateral geniculate nucleus (dLGN) where it is relayed to the primary visual cortex (V1). A large number of axons for retinal ganglion cells (RGCs), however, also terminate in the superior colliculus. As much as 85–90% of the RGCs project to the superficial layers of the superior colliculus (sSC) in the mouse[1]. The mouse sSC shows feature selectivity for orientation and direction[2–4], in part inherited from the retina[5]. It also receives input from V1[6–8]. Through these connections, V1 influences responses in the sSC in awake mice[9,10]. The superior colliculus projects to the visual thalamus and could in turn influence processing in the visual cortex. Indeed, lesioning of the superior colliculus affects velocity tuning in higher visual cortical areas[11]. This effect is thought to be mediated by the strong connection from the superior colliculus to the lateral posterior (LP) nuclei, homologous to the primate pulvinar areas, in the visual thalamus. The superior colliculus could potentially also influence the input to the primary visual cortex through its projection to the shell of the dLGN[12]. This tectogeniculate projection is anatomically present in all studied mammals[13], but there has been no evidence that it has a functional effect on thalamocortical visual processing. Superior colliculus, and its non-mammalian homolog the optic tectum, is a key area in the attentional network of vertebrates[14,15]. The basis for its role in attention may be its representation of a saliency map[16,17]. In this map, higher saliency corresponds to a higher firing rate. Here, we wanted to investigate if the computation of such a saliency map in the superior colliculus could influence neural responses in the primary visual cortex, even in a situation where visual attention does not play any role. Combining electrophysiology with optogenetic and pharmacologic approaches, we have examined the influence of the superior colliculus on visual responses in V1. We found that optogenetically silencing superior colliculus reduced response in V1 to optimally sized stimuli. Surprisingly, this effect is not via the strong SC to LP pathway, but via the anatomically smaller tectogeniculate pathway. A size-independent gain modulation of the dLGN by the superior colliculus is transformed into a size-dependent gain modulation in V1.

## Results

**Functional modulation of V1 responses by sSC.** To look at the influence of sSC on V1, we reduced activity in the excitatory neurons in the sSC by optogenetically activating the inhibitory neurons. To achieve this, we expressed channelrhodopsin-2[18] (ChR2) in sSC inhibitory neurons through injecting a Cre-dependent virus in the sSC of Gad2-Cre driver line mice[19]. We simultaneously recorded with laminar micro-electrodes in the sSC and V1 of anesthetized mice, while a laser-coupled light fiber was kept above sSC (Fig. 1a, b). Care was taken to have similar receptive field (RF) positions in the sSC and V1 recordings, which were in front of the animal (mean distance from the main body axis sSC: $6.6 \pm 7.2$ deg vs V1: $17.9 \pm 6.5$ deg, mean ± s.e.m.; 14 mice, 38 sSC RFs and 45 V1 RFs, Fig. 1c). Optogenetic activation of the inhibitory network of *Gad2*-positive neurons by blue light decreased the visual response in the sSC on average by 33% (laser off: $17.4 \pm 1.2$ Hz vs laser on: $11.6 \pm 0.9$ Hz, $p < 0.00001$, Wilcoxon test, 14 mice, 182 units, Fig. 1d). We measured the visual response in V1 to disks with drifting gratings of different sizes centered at the V1 RFs, while we optogenetically inhibited the sSC every other trial. The effect of sSC silencing depended on the size of the stimulus ($p = 0.0005$, two-way analysis of variance (ANOVA), Fig. 1e, f). The response to optimally sized stimuli was reduced by suppressing sSC activity (laser off: $30.5 \pm 2.2$ Hz vs laser on: $27.8 \pm 2.1$ Hz; $p = 4.7 \times 10^{-14}$, Wilcoxon test; 14 mice, 160 units, Fig. 1g), but there was no significant decrease in the visual

response to the largest size visual stimulus (laser off: $20.0 \pm 1.7$ Hz vs laser on: $19.7 \pm 1.7$ Hz; $p = 0.06$, Wilcoxon test, Supplementary Fig. 1). On average, only V1 responses to near optimal size stimuli were decreased by optogenetic inhibition of the sSC (3-size steps below optimal: $p = 0.17$, 2-size steps: $p = 4.7 \times 10^{-5}$, 1-step: $p = 4.9 \times 10^{-7}$, optimal: $p = 1.5 \times 10^{-14}$, 1-step above: $p = 4.4 \times 10^{-5}$, 2-steps: $p = 0.02$, 3-steps: $p = 0.06$, 4-steps: $p = 0.56$, Wilcoxon test, Fig. 1f). Spontaneous activity in V1 was not reduced by inhibiting the sSC (laser off: $3.77 \pm 0.34$ Hz, laser on: $4.14 \pm 0.39$, mean ± s.e. m., $p = 0.37$, Wilcoxon test, 14 mice, 160 units). The effects were very similar when mice were awake, with again a clear difference in effect depending on the size of the stimuli ($p = 0.0039$, two-way ANOVA, 5 mice, 64 units), caused by a strong reduction (18%) in response to optimal size stimuli (laser off: $67.1 \pm 4.8$ Hz vs laser on: $55.1 \pm 4.0$ Hz, $p = 3.5 \times 10^{-12}$, Wilcoxon test, Supplementary Fig. 2A-B) and a much weaker reduction (11%) for the largest size (laser off: $46.7 \pm 3.5$ Hz vs laser on: $41.3 \pm 3.1$ Hz; $p = 4.1 \times 10^{-12}$, Wilcoxon test; Supplementary Fig. 2C).

The effect of sSC on V1 activity thus appeared to be different from an overall gain change. The differential effect of inhibiting sSC for different size stimuli becomes clearly apparent if we compute the amount of surround suppression in V1, quantified by a surround suppression index (SSI, see Methods), without and with suppression of sSC activity (anesthetized SSI, laser off: $0.389 \pm 0.016$ vs laser on: $0.318 \pm 0.017$, mean ± s.e.m., $p < 0.00001$, Wilcoxon test, 14 mice, 160 units, Fig. 1h). The size dependence was also evident in the reduction of awake V1 SSI (SSI laser off: $0.31 \pm 0.02$ vs laser on: $0.26 \pm 0.02$; $p = 1.4 \times 10^{-12}$, Wilcoxon test, Supplementary Fig. 2D). The surround suppression decrease in the V1 units was correlated to the amount of sSC suppression (Pearson's $r = 0.95$, $p = 0.0004$, non-zero slope test, Supplementary Fig. 3). The effect of sSC modulation was seen across the entire depth of V1. The change in surround suppression was significant over all depths of V1 (top channels: $p = 1.2 \times 10^{-6}$, middle: $p = 7.4 \times 10^{-5}$, deep: $p = 0.00014$, Wilcoxon test, Fig. 1i) and was not significantly different between V1 depths (top vs middle channels: $p = 0.54$, top vs deep: $p = 0.33$, middle vs deep: $p = 0.26$, Mann–Whitney $U$-tests, Fig. 1i). The modulation was not through a direct effect of the laser light on the tissue or retina, as control experiments with the same light in mice without ChR2 expression did not show any effect on the response to the optimal size or the SSI of V1 units (response, laser off: $30.5 \pm 3.8$ Hz vs laser on: $30.2 \pm 3.7$ Hz, $p = 0.33$; SSI, laser off: $0.374 \pm 0.029$ vs laser on: $0.377 \pm 0.029$; $p = 0.76$, Wilcoxon test; 5 mice, 52 units, Fig. 1j, k).

**Tectal influence on V1 size tuning is not mediated by LP.** We thus found an influence of sSC on V1, but there is no direct projection from the sSC to V1. There is an indirect projection from the sSC to V1 via LP (tectopulvinar pathway) and via the dLGN (tectogeniculate pathway). Compared to the dLGN, LP is more densely innervated by the sSC. We first assessed the impact of this connection on LP responses. We again optogenetically suppressed sSC activity in anesthetized mice, but now while recording from LP (Fig. 2a). The placement of the electrode in LP was verified post hoc using histology (Fig. 2b). Visual responses in LP were strongly reduced when sSC was suppressed (laser off: $9.8 \pm 2.5$ Hz vs laser on: $3.3 \pm 1.1$ Hz, $p = 0.0005$, Wilcoxon test; 4 mice, 12 units; Fig. 2c). Next, we optogenetically suppressed sSC activity but now while also recording from V1, before and after silencing LP by an injection of fluorescent muscimol in LP (Fig. 2d). This way we could determine whether silencing LP removes the influence of sSC on V1 responses. To make sure that we silenced LP, we also recorded in LP, before and after the muscimol injection. The RFs in all these LP, sSC, and V1

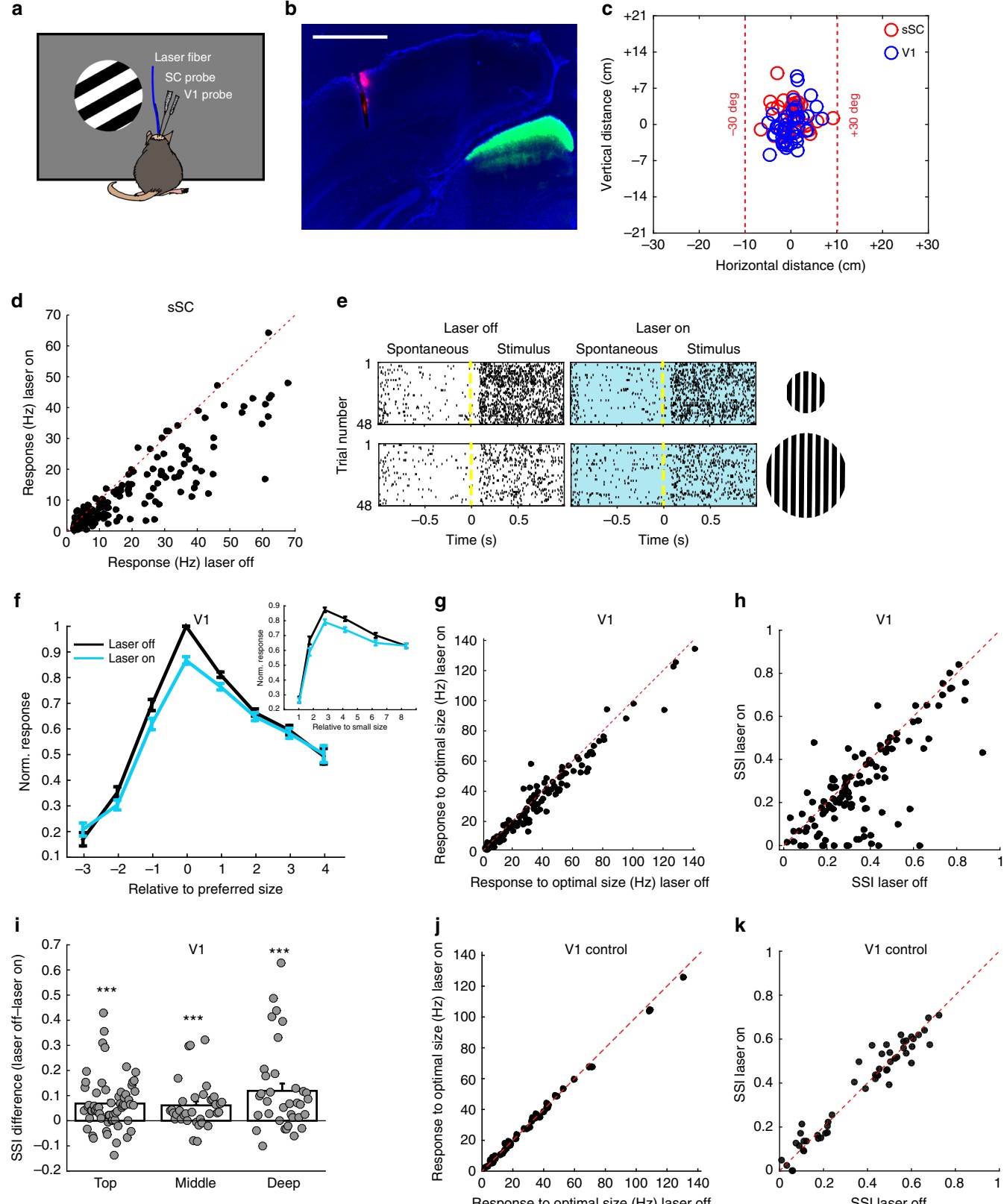

recordings were in front of the mouse, within ±30 degrees from the main body axis, and retinotopically matched (Fig. 2e). As has been reported before, LP RFs were large[20]. It is difficult to inject muscimol in LP without leakage into the neighboring dLGN. Therefore, we used fluorescence-conjugated muscimol in order to

determine the spread of the muscimol after each experiment and reach the optimum location and amount of the injection. Figure 2f shows an example of the muscimol diffusion in LP. The recordings from the LP by laminar probes showed the high efficiency of the muscimol in silencing the LP (before muscimol:

**Fig. 1** Optogenetically inhibiting the sSC reduces surround suppression in V1. **a** Laminar recording electrodes in the sSC and V1 of an anesthetized Gad2-Cre mouse, expressing channelrhodopsin-2 in the sSC, with a laser-coupled fiber placed above the sSC. Screen was positioned 17.5 cm in front of the mouse. **b** Coronal slice showing ChR2-eYFP expression (green) in the sSC and the DiI (red) trace of the V1 recording electrode. DAPI in blue. Scale bar is 1 mm. **c** Distribution of recorded receptive field locations of 38 sSC and 45 V1 units (14 mice). Receptive field sizes are not indicated. **d** Responses to full screen gratings in the sSC were reduced by optogenetic activation of the Gad2-cells ($p < 0.00001$, Wilcoxon test, 14 mice, 182 units). **e** Spike rastergrams from an example V1 unit for optimal size and large size gratings, without and with (blue background) optogenetic inhibition of the sSC. Each line in the rastergram represents one trial. **f** Population average size tuning in V1 without and with optogenetic inhibition of sSC. Horizontal axis shows size step relatively to the optimal size per unit. Inset shows size step relative to the smallest size presented within a recording. Error bars represent s.e.m. **g** Responses to optimal size grating in V1, without and with laser light on sSC ($p = 4.7 \times 10^{-14}$, Wilcoxon test; 14 mice, 160 units). **h** Surround suppression index in V1 was reduced by optogenetically inhibiting the sSC ($p < 0.00001$, Wilcoxon test). **i** Surround suppression in V1 was reduced in all depths by optogenetically inhibiting the sSC ($p < 0.001$, Wilcoxon tests). There is no significant difference in change between depths ($p > 0.25$, Mann–Whitney U-tests). Top, middle, deep correspond to the top, middle, and bottom third of the visually responsive channels on the electrode spanning V1; ***$p < 0.001$. **j** Responses to optimal size grating in V1 in wild-type control mice was not changed by laser light on sSC ($p = 0.33$, Wilcoxon test; 5 mice, 52 units). **k** Surround suppression in V1 was not changed by laser light in wild-type control mice ($p = 0.76$, Wilcoxon test)

$7.51 \pm 1.30$ Hz vs after: $0.25 \pm 0.03$ Hz, $p = 8.3 \times 10^{-6}$, Wilcoxon test; 5 mice, 26 units; Fig. 2g). Silencing LP did not change overall V1 responses to optimal or largest size stimuli in these experiments (optimal size, before muscimol: $17.4 \pm 2.3$ Hz vs after: $16.2 \pm 2.1$ Hz, $p = 0.70$; largest size, before muscimol: $9.1 \pm 1.3$ Hz vs after: $8.8 \pm 1.2$ Hz, $p = 0.62$, Mann–Whitney U-tests; 5 mice, 42 units before, 49 different units after; effect of muscimol: $p = 0.23$, interaction of muscimol and size: $p = 0.99$, two-way ANOVA, Fig. 2h, i) and in a separate set of experiments where we recorded the same V1 neurons before and after silencing LP (optimal size, before muscimol: $57.2 \pm 6.5$ Hz vs after: $51.3 \pm 5.3$ Hz, $p = 0.11$; large size, before muscimol: $48.0 \pm 5.7$ Hz vs after: $43.7 \pm 4.8$ Hz, $p = 0.24$, Wilcoxon test; 4 mice, 55 units; effect of muscimol: $p = 0.55$, interaction of muscimol and size: $p = 0.95$, two-way ANOVA, Supplementary Fig. 4A-F). The difference in significance of the overall effect of LP silencing compared to sSC silencing was not due to a difference in the number of units (Supplementary Fig. 4G). The selective reduction of the responses to optimally sized stimuli by the optogenetic inhibition of the sSC remained after LP silencing (Fig. 2j). This was apparent by the similar optogenetic effects on SSI before and after muscimol (before muscimol: SSI laser off: $0.473 \pm 0.042$ vs laser on $0.403 \pm 0.026$, $p = 9.3 \times 10^{-9}$, paired t-test; 5 mice, 42 units; Fig. 2k; after muscimol: laser off: $0.482 \pm 0.025$ vs laser on: $0.412 \pm 0.020$; $p = 3.4 \times 10^{-5}$, paired t-test; 5 mice, 49 units; Fig. 2l). The amount of SSI reduction in V1 was thus unaffected by silencing LP (before muscimol: $14.5 \pm 2.4\%$ vs after: $13.9 \pm 3.1\%$, $p = 0.92$, Mann–Whitney U-test; Fig. 2m). This result shows that the size-specific sSC modulation of the V1 that we found in the anesthetized mouse is not through the tectopulvinar pathway. Moreover, in order to see if LP has any effect on the V1 activity and/or V1 SSI, we also used another strategy. The calcium-binding protein Calretinin (coded by the *Calb2* gene) is expressed in many excitatory cells of the LP[21] and not at all in the dLGN (Supplementary Fig. 5A). Therefore, we injected a virus with a Cre-dependent archaerhodopsin vector[22] in the LP of Calb2-Cre mice to optogenetically reduce LP activity (Supplementary Fig. 5B-C). While LP activity was 22% reduced by laser light (laser off: $18.0 \pm 2.6$ Hz vs laser on: $14.0 \pm 2.4$ Hz, $p = 0.003$, paired t-test; 2 mice, 7 units; Supplementary Fig. 5D), there was no effect on V1 responses (optimal size, laser off: $22.4 \pm 5.0$ Hz vs laser on: $22.7 \pm 5.2$ Hz, $p = 0.84$, Wilcoxon test; 2 mice, 13 units; optogenetic effect: $p = 0.51$, interaction of optogenetics and size: $p = 0.80$, two-way ANOVA, Supplementary Fig. 5E-F) and also no change in V1 surround suppression (SSI, laser off: $0.38 \pm 0.04$ vs laser on: $0.41 \pm 0.04$, $p = 0.22$, paired t-test; 2 mice, 13 units; Supplementary Fig. 5G).

**Reducing sSC activity reduces visual response in dLGN.** The other disynaptic pathway connecting sSC to V1 is via the dLGN.

Earlier work indicated that a large fraction of dLGN neurons receive inhibitory input from the mouse sSC[23]. However, later work found less than 5% of retrogradely cholera toxin B subunit (CTB)-labeled neurons to be GABAergic[12]. To get an independent estimate of the fraction of the inhibitory sSC neurons projecting to the dLGN, we injected a retrograde canine adenovirus type 2 (CAV2) virus in the dLGN to express ZsGreen in neurons projecting to the injection location. We made the injections in mice expressing tdTomato in all GABAergic cells (Fig. 3a). All of the 28 cells retrogradely labeled in the sSC were non-GABAergic. A crude upper bound of the fraction of GABAergic cells of the cells projecting from the sSC to the dLGN is thus less than 1 in 28, i.e., 4%. We therefore expect that the activation of GABAergic cells in the sSC primarily results in a reduction of glutamatergic input from the sSC to the dLGN. Injection of two adeno-associated viruses (AAV) for expressing different fluorescent markers in the sSC showed that the projection from sSC to dLGN is retinotopically organized, but clearly smaller than the projection to LP (Fig. 3b). To determine if the dLGN mediates the V1 effect of optogenetically inhibiting the sSC, it is not informative to silence dLGN, because it would also silence V1. Instead, we examined the dLGN responses, while we again optogenetically reduced activity in the sSC (Fig. 3c, d). RFs were matched and within ±30 degrees from the central body axis (Fig. 3e). Suppressing sSC activity reduced visual responses of dLGN neurons ($p = 7.0 \times 10^{-6}$ two-way ANOVA, 6 mice, 26 units; Fig. 3f), in equal proportion for all tested stimulus sizes ($p = 0.95$, interaction between optogenetics and size in two-way ANOVA). Indeed, there was no change in the ratio of optimal to large size stimuli, as surround suppression in the dLGN was not changed (SSI laser off: $0.251 \pm 0.053$ vs laser on: $0.217 \pm 0.069$, $p = 0.95$; Wilcoxon test; 6 mice, 26 units; Fig. 3g). Reducing sSC activity thus reduced the overall gain in the dLGN, rather than a specific reduction of responses to optimally sized stimuli. The spontaneous activity in the dLGN was not changed (laser off: $7.9 \pm 1.0$ Hz, laser on: $8.3 \pm 1.2$ Hz, $p = 0.38$, Wilcoxon test). The visual response reduction was significant at the most dorsal 150 μm of the dLGN, presumably corresponding to the dLGN shell (large size response, laser off: $10.3 \pm 1.5$ Hz vs laser on: $7.2 \pm 1.7$ Hz; $p = 0.004$, paired t-test; 6 mice, 18 units; Fig. 3h) and was not significant at the more ventral part of the dLGN (laser off: $15.6 \pm 2.4$ Hz vs laser on: $15.3 \pm 2.5$ Hz; $p = 0.59$, paired t-test; 6 mice, 19 units; Fig. 3h).

The shell of the dLGN is the recipient of the input from the sSC[12]. The shell also receives input from direction-selective retinal ganglion cells[24] and has a higher proportion of direction and orientation selective neurons than the core[25]. This direction and orientation selectivity is transferred to V1[26,27]. For this reason, we also studied the effect of suppressing sSC activity on V1 orientation and direction selectivity by showing full screen gratings drifting in different directions. There was no significant

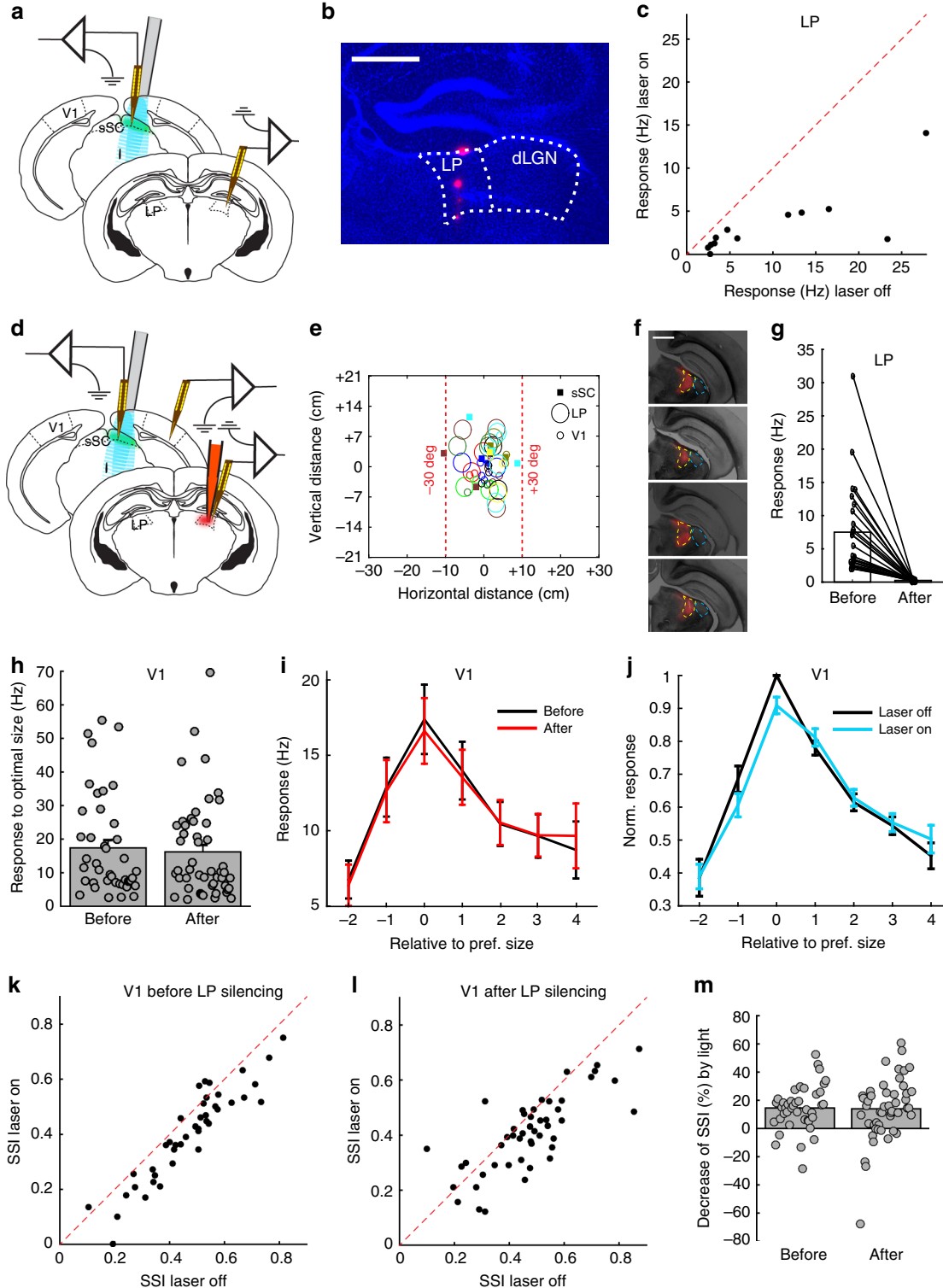

change, however, in the direction selectivity index when we include all cells (laser off: 0.0861 ± 0.0094 vs laser on: 0.0834 ± 0.0099, mean ± s.e.m., *p* = 0.89, Wilcoxon test; 15 mice, 38 single units, Supplementary Fig. 6A) or when we only include direction-sensitive cells (cells with direction selectivity index (DSI) > 0.1, *p* = 0.98, Wilcoxon test; 15 mice, 18 single units). Also, the orientation selectivity in V1 was not significantly changed by optogenetic inhibition of sSC (laser off: 0.202 ± 0.024 vs laser on: 0.209 ± ± 0.023, *p* = 0.39, Wilcoxon test; 15 mice, 38 single units,

Supplementary Fig. 6B) when we include all cells, or only the orientation-sensitive cells (cells with orientation selectivity index (OSI) > 0.2, *p* = 0.52, Wilcoxon test; 15 mice, 15 single units).

**V1 modulation is through the direct tectogeniculate pathway.** The retrograde CAV2 injections also suggested a potential alternative route from the sSC to the dLGN. We found cells in the contra- and ipsilateral parabigeminal nucleus (PBG) that project to the dLGN (Supplementary Fig. 7). The PBG is a small

**Fig. 2** Effect of sSC on V1 surround suppression is not mediated by LP. **a** Laminar recording electrodes in LP and the sSC of anesthetized Gad2-cre mouse, expressing channelrhodopsin-2 in the sSC, with a laser-coupled fiber placed above the sSC. Screen was positioned in front of the mouse. **b** Coronal slice showing a DiI trace of the recording electrode in LP. DAPI in blue. Scale bar is 0.5 mm. **c** Optogenetic inhibition of sSC strongly reduced visual responses to full screen gratings in LP ($p = 0.0005$, Wilcoxon test, 4 mice, 12 units). **d** Laminar recording electrodes in V1 and the sSC, and in LP before and after injection of fluorescent muscimol in LP. **e** Receptive field centers of recorded units in sSC, LP and V1 in this set of experiments. Shapes indicate brain areas. Colors represent the different experiments. Receptive field sizes are not indicated. **f** Example of one fluorescence-conjugated muscimol injection shown in a few coronal slices. LP outline is indicated by yellow dashed line. **g** Muscimol effectively silenced LP (5 mice, 26 units). **h** Response to optimal size in V1 was not reduced by LP silencing ($p = 0.70$, Wilcoxon test; 5 mice, 42 units before, 49 different units after). Bars represent mean. **i** Size tuning in V1 was unchanged by LP silencing. Error bars represent mean ± s.e.m. **j** Optogenetic suppression of sSC by laser continues to suppress optimal sized stimuli after LP silencing. **k** V1 SSI was reduced by optogenetic inhibition of sSC, before LP silencing ($p = 9.3 \times 10^{-9}$, paired $t$-test; 5 mice, 42 units). **l** V1 SSI was reduced by optogenetic inhibition of sSC, also after LP silencing ($p = 3.4 \times 10^{-5}$, paired $t$-test; 5 mice, 49 units). **m** No difference in the amount of SSI reduction by optogenetic inhibition of the sSC, before and after LP silencing ($p = 0.92$, Mann–Whitney $U$-test). Bars represent mean

cholinergic midbrain nucleus that receives its main input from the sSC[6]. It could thus be that the sSC effect on V1 was through the sSC–PBG–dLGN pathway and not through the direct sSC–dLGN pathway. Guided by stereotactic coordinates and auditory and visual responses, we recorded in the PBG (Fig. 4a, see Methods for targeting details). Figure 4b, c shows an example microelectrode trace and visual response in the PBG. As expected, optogenetically inhibiting the sSC strongly reduced visual responses in the PBG (laser off: 9.5 ± 1.6 Hz vs laser on: 3.7 ± 1.0 Hz, $p = 4 \times 10^{-5}$; Wilcoxon test; 4 mice, 22 units; Fig. 4d). To check whether the sSC modulation of V1 was mediated by the PBG, we silenced the PBG with an injection of fluorescent muscimol in wild-type animals while recording in the PBG and V1 (Fig. 4e). The large RFs of the PBG had a high overlap with the RFs of V1 (Fig. 4f). Figure 4g shows an example of the spread of the fluorescent muscimol covering the PBG. Electrophysiological recordings confirmed that the PBG was silenced (before muscimol: 7.37 ± 1.30 Hz vs after: 0.06 ± 0.22 Hz, $p = 0.002$, Wilcoxon test; 4 mice, 10 units; Fig. 4h). Figure 4i shows V1 responses to different visual stimuli sizes, normalized by peak responses before silencing the PBG. There was no difference in V1 responses before and after PBG silencing ($p = 0.57$, two-way ANOVA). There was also no change in the ratio of responses to optimal, small, and large size stimuli (optimal size, before muscimol: 45.3 ± 6.1 Hz vs after: 42.2 ± 5.1 Hz, $p = 0.22$, Wilcoxon test; 4 mice, 43 units; small size, before muscimol: 6.8±1.6 Hz vs after: 6.2 ± 1.4 Hz, $p = 0.10$; large size, before muscimol: 35.2 ± 4.8 Hz vs after: 31.9 ± 3.7 Hz, $p = 0.12$; Supplementary Fig. 8A-C), as surround suppression in V1 was also not affected (SSI before muscimol: 0.233 ± 0.018 vs after: 0.242 ± 0.015, $p = 0.27$, paired $t$-test; 4 mice, 43 units, Fig. 4j). The difference in significance of the overall effect of PBG silencing compared to sSC silencing was not due to a difference in the number of units (Supplementary Fig. 8D). Therefore, the effect of the sSC on V1 responses in the anesthetized animal was not through the indirect pathway via the PBG to the dLGN, but directly through the dLGN.

**Gain modulation in dLGN by the sSC changes V1 size tuning.** Optogenetic inhibition of the sSC caused a proportionally equal decrease of the dLGN responses to the visual stimuli of the different sizes. At first sight, this does not match the selective reduction in V1 response to optimally sized stimuli. We wondered whether this selective reduction and the consequential reduction in cortical surround suppression could be an indirect effect of a lower gain in dLGN. The effect of reducing sSC activity on the dLGN appeared similar to a reduction of stimulus contrast. We wanted to see if a reduction in stimulus contrast that reduced cortical responses to optimal sized stimuli to the same amount as optogenetic inhibition of the sSC would have a similar effect on cortical surround suppression. Therefore, we measured in V1 the contrast response curve for the stimulus size that was

optimal for the largest population of V1 neurons, with and without optical inhibition of sSC. We then chose the highest contrast at which the laser caused a reduction of about 15% in response (Fig. 5a). From the contrast response curve in the laser-off condition, we picked a lower contrast which gave approximately the same amount of response reduction as optogenetically inhibiting the sSC. At this lower contrast, we measured the tuning curve again (Fig. 5b). Not surprisingly, the response in V1 at this lower contrast is reduced for optimally sized stimuli (Fig. 5c). The reduction, however, is much weaker for the largest stimuli ($p = 6.5 \times 10^{-25}$, Kruskal–Wallis test; 5 mice, 63 units; Fig. 5c). This is reflected in a much lower surround suppression in V1 for these lower contrast stimuli than for the high contrast stimuli (high contrast SSI: 0.197 ± 0.017 vs lower contrast: 0.147 ± 0.016, $p = 3.4 \times 10^{-7}$, Wilcoxon test, Fig. 5d), similar to the effect of optogenetically inhibiting the sSC. The population size tuning curve measured at the lower contrast with an active sSC resembled the size tuning curve measured at high contrast and a suppressed sSC (Fig. 5c). Optogenetically inhibiting sSC activity reduced the surround suppression for the lower contrast stimuli even further (Fig. 5e). Figure 5f shows that inactivating sSC and lowering the contrast both decreased V1 SSI to a similar extent (change by inactivating sSC, $p = 0.00019$; change by lowering contrast $p = 0.041$; difference between inactivated sSC condition and lower contrast: $p = 0.19$, all Bonferroni-corrected Mann–Whitney $U$-tests). The change in size tuning in V1 by optogenetically inhibiting sSC induced can thus be explained by a size-independent gain reduction in the dLGN.

## Discussion

We investigated the influence of superior colliculus on the visual responses in the mouse primary visual cortex. We found that suppressing superior colliculus activity reduced the gain in the dLGN. The reduced dLGN response led to a reduction of responses in V1, except for very large stimuli, effectively reducing cortical surround suppression. This response change was similar to the effect of a reduction in contrast. The LP/Pulvinar did not play a role in this collicular modulation of V1 responses.

Previous studies have primarily studied the influence of superior colliculus on extrastriate cortex. In macaque, silencing the intermediate and deep superior colliculus had only a weak effect on responses of extrastriate visual cortex[28], but removing or lesioning the whole superior colliculus had a strong effect when V1 was already inactivated[29,30]. Superior colliculus inactivation lead to a loss of response to high stimulus velocities in higher order visual cortical areas of cat and mouse[11,31]. In mouse, lesioning superior colliculus reduced response in V1 across velocities[11]. The visual stimuli used for this latter study were about 40 degrees in diameter. This agrees with our finding that in our range of intermediate stimulus sizes, optogenetically inhibiting sSC reduces V1 response.

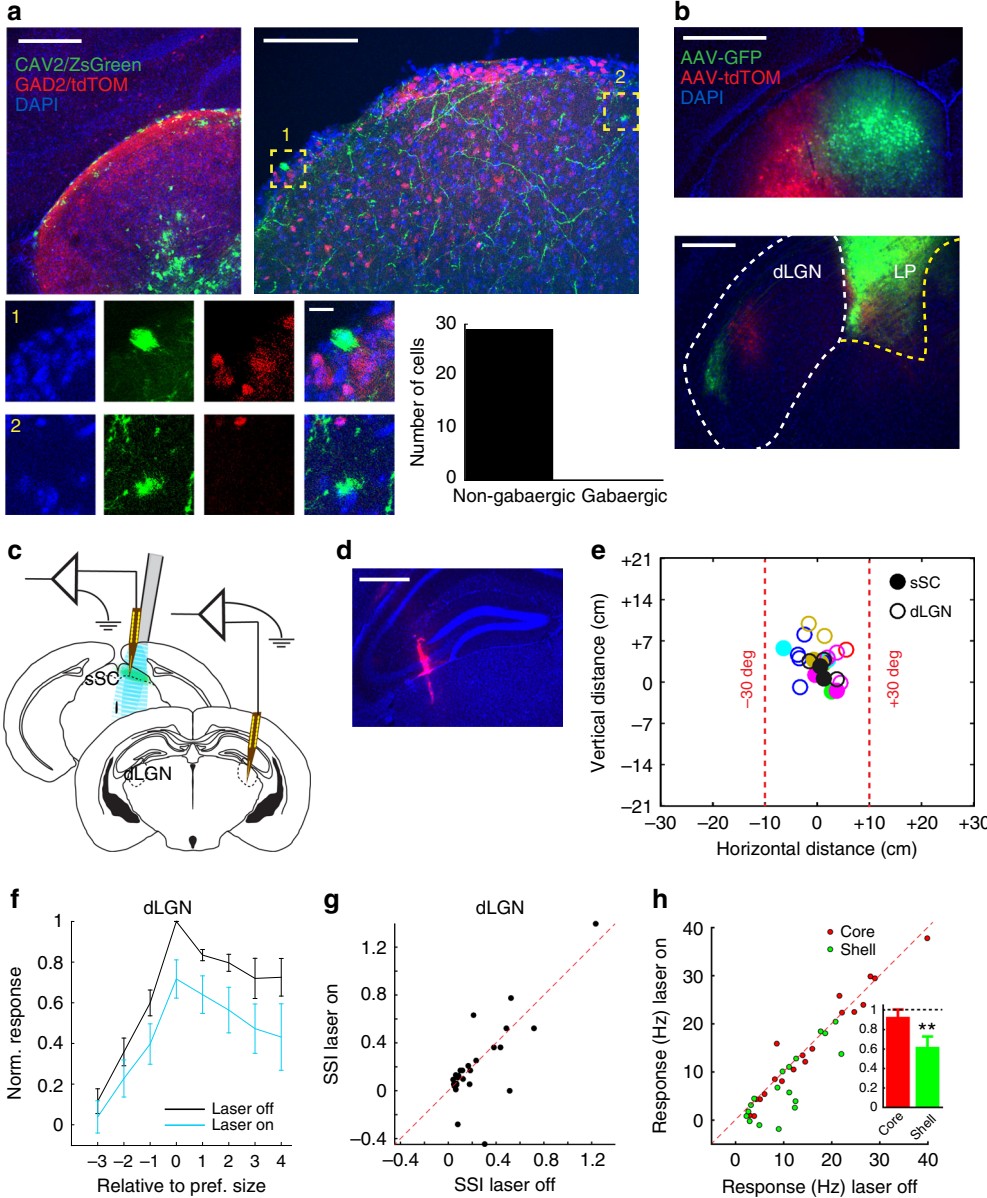

**Fig. 3** The sSC increases dLGN gain via excitatory projection. **a** Top left, coronal slice of dLGN of a Gad2 x Ai14 tdTomato reporter mouse transfected with a retrograde CAV2 virus with a ZsGreen expressing vector. Scale bar is 205 μm. Top right, coronal slice of sSC in the same experiment showing retrogradely labeled neurons (green) and inhibitory neurons (red). Scale bar is 100 μm. Bottom left, 1 and 2 show magnifications of labeled cells indicated at top right. Scale bar is 10 μm. Bottom right, all 28 labeled cells were Gad2-Cre negative. **b** Top, coronal slice of SC injected medially with AAV expressing GFP and more laterally with AAV expressing tdTomato. Scale bar is 500 μm. Bottom, coronal slice from the same experiment showing sSC innervation of the dLGN. Dashed lines show outline of dLGN (left) and LP (right). Scale bar is 250 μm. **c** Laminar recording electrodes in the dLGN and sSC of anesthetized Gad2-Cre mouse, expressing channelrhodopsin-2 in the sSC, with a laser-coupled fiber placed above the sSC. **d** Example trace of DiI (red) left in the dLGN by recording electrode. DAPI in blue. Scale bar is 500 μm. **e** Distribution of receptive field centers for this set of experiments. Colors indicate the different experiments. Receptive field sizes are not indicated. **f** Optogenetic inhibition of sSC reduced visual responses in the dLGN (optogenetic effect $p = 7.0 \times 10^{-6}$, interaction of optogenetics and size $p = 0.95$, two-way ANOVA, 6 mice, 26 units). Size is the rank of the stimulus size in a recording, counted from the preferred stimulus in that recording. Error bars indicate s.e.m. **g** Surround suppression in dLGN was unchanged by optogenetic inhibition of sSC ($p = 0.95$, Wilcoxon test, 6 mice, 26 units). **h** Optogenetic inhibition reduced responses to the largest size gratings in the dorsal (shell) part of the dLGN ($p = 0.004$, paired $t$-test, 6 mice, 18 units) and not the ventral (core) part ($p = 0.59$, paired $t$-test, 6 mice, 19 units). Inset shows mean and s.e.m. of the response when the laser was on, relative to when it was off

It has long been known that across species superior colliculus projects to the dLGN[13], but the contribution of this tectogeniculate pathway to dLGN responses has remained unknown[6]. We confirmed that the large majority of the cells in the sSC projecting to the dLGN are not GABAergic[12], although some inhibitory neurons project to the dLGN[23]. Mouse sSC axons terminate mostly in the dLGN shell[12]. In agreement with this, we found that

only the dorsal channels in our dLGN recordings were modulated by suppressing the sSC.

The shell of the dLGN receives direct input from different types of direction-selective retinal ganglion cells[24,32,33] and contains a higher proportion of orientation- or direction-selective relay cells than the dLGN core[25]. The shell also has a high number of morphologically W-like cells[34] that receive retinal and

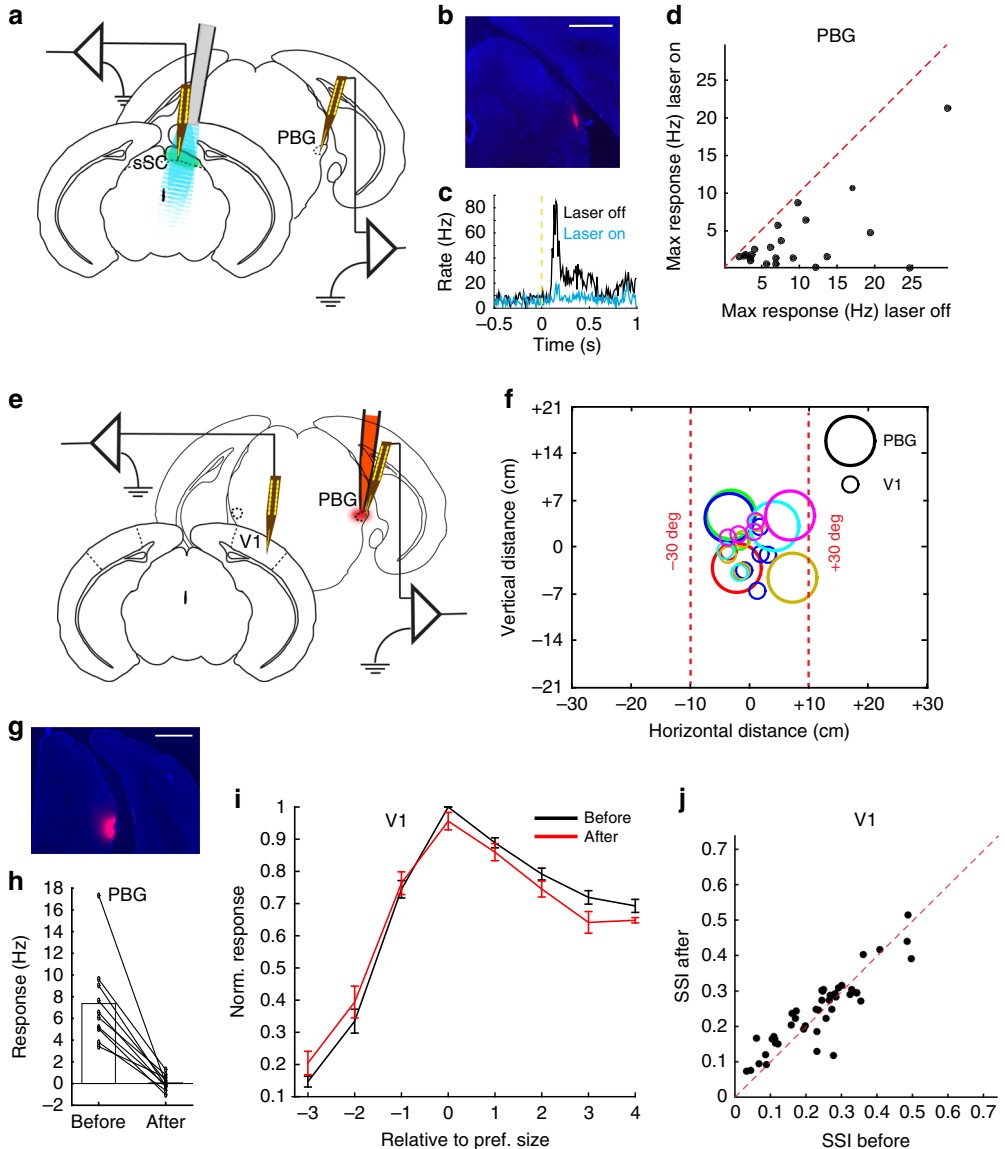

**Fig. 4** Effect of sSC on V1 is not mediated by PBG. **a** Laminar recording electrode in the PBG of anesthetized Gad2-cre mouse, expressing channelrhodopsin-2 in the sSC, with a laser-coupled fiber placed above the sSC. **b** Coronal slice showing trace of DiI (red) from the recording electrode in PBG. DAPI in blue. Scale bar is 0.5 mm. **c** Example peristimulus time spike histogram in the PBG, without and with optogenetic inhibition of the sSC. **d** Maximum response to full screen gratings in PBG was reduced by optogenetic inhibition of the sSC ($p = 4 \times 10^{-5}$, Wilcoxon test, 4 mice, 22 units). **e** Laminar recording electrodes in V1 and the PBG, before and after injection of fluorescent muscimol in the PBG, in the anesthetized mouse. **f** Receptive field centers of recorded units in V1 and the PBG for this set of experiments. Colors represent the different experiments. Receptive field sizes are not indicated. **g** Coronal slice showing fluorescent muscimol in the PBG. DAPI in blue. Scale bar is 0.5 mm. **h** Muscimol silenced the PBG ($p = 0.002$, Wilcoxon test; 4 mice, 10 units). **i** Visual responses in V1 were not changed by PBG silencing ($p = 0.57$, two-way ANOVA; 4 mice, 43 units). Size is the rank of the stimulus size in a recording, counted from the preferred stimulus in that recording. Error bars represent mean ± s.e.m. **j** Surround suppression in V1 was not changed by PBG silencing ($p = 0.27$, paired $t$-test; 4 mice, 43 units)

tectal input[12]. Rodent W-like cells are likely to be homologous to carnivore W and primate koniocellular cells[35]. The preponderance of W-like cells and its connectivity pattern suggests a homology of the mouse shell to the carnivore C-layers and primate koniocellular layers[24]. The relay neurons of the shell preferentially target the superficial layers of V1 and transmit this direction and orientation selectivity[26,27]. Surprisingly, however, we found neither changes in orientation or direction selectivity in V1 nor a larger effect in superficial V1. The functional effect of the superior colliculus on dLGN that we found is modulatory rather than driving, which matches the relatively weak anatomical connection, although the synapses also possess characteristics that

define drivers[12]. The collicular effect on dLGN shell neurons is modulating visual responses in V1 regardless of feature preference or depth.

Another possibility that we considered for explaining the modulation of geniculate neurons is a pathway going through the PBG, a cholinergic area homologous to the nucleus isthmii pars parvocellularis in non-mammalian vertebrates. The PBG is visually responsive, and gets visual input via a topographic projection of the ipsilateral sSC[6,36]. The PBG projects primarily back to the ipsi- and contralateral SC, and also to the dLGN. In rats, this projection is primarily to the contralateral dLGN[37], but this is species dependent. In the cats the projection is bilateral, and in

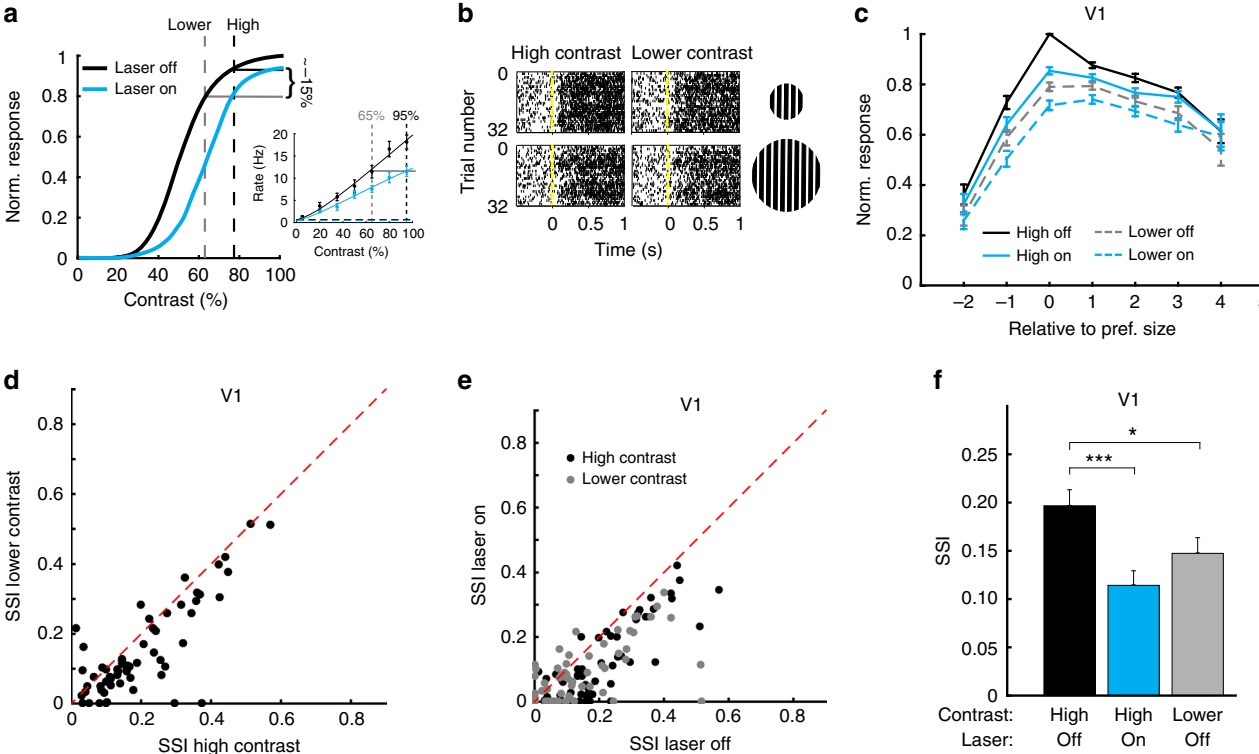

**Fig. 5** Lowering contrast reduces V1 surround suppression like optogenetically inhibiting sSC. **a** Contrast tuning in V1 was measured for an optimal size stimulus, without (black) and with (blue) optical inhibition of the sSC. The highest contrast at which the population response showed a reduction of approximately 15% was set as high contrast level. The contrast level at which the population gave a roughly equal response when the sSC is not inhibited was set as lower contrast level. Inset, example contrast tuning curve for one V1 unit. **b** Example rastergram of V1 neuron for gratings shown at two sizes at high (90%) and lower (70%) contrast. Different lines show different trials with different drifting directions. Stimulus started at time 0. **c** Population average size tuning curves for lower contrast resembled the size tuning curve at high contrast with optogenetic inhibition of the sSC. All responses were normalized to the response of the preferred stimulus at high contrast, without optogenetic inhibition of the sSC. Error bars indicate s.e.m. **d** Surround suppression was lower at lower contrast ($p = 3.4 \times 10^{-7}$, Wilcoxon test; 5 mice, 63 units). **e** Optogenetic inhibition of the sSC reduced V1 surround suppression for high and lower contrasts. **f** Surround suppression for high contrast without and with optogenetic inhibition of sSC and for lower contrast without optogenetic inhibition of the sSC (change by inactivating sSC, $p = 0.00019$; change by lowering contrast $p = 0.041$; difference between inactivated sSC condition and lower contrast: $p = 0.19$, all Bonferroni-corrected Mann–Whitney $U$-tests). Bars indicate mean and s.e.m.; *$p < 0.05$, ***$p < 0.001$

the primates it is primarily ipsilateral[38]. Our injections of retrogradely labeling virus in the dLGN showed projecting cells in both ipsi- and contralateral PBG. Silencing the PBG, however, did not change surround suppression in ipsilateral V1. This excludes the SC–PBG–dLGN pathway as an option for conveying the collicular influence on V1 surround suppression.

The pathway previously thought to be responsible for changes in visual responses in the cerebral cortex following silencing superior colliculus passes the LP/pulvinar nuclei[11]. There is a large projection from the SC to pulvinar in all mammals[6]. This projection comes from a cell type that is distinct from the dLGN-projecting sSC neurons[39]. It could be a driving input[40], but SC lesions have little impact on activity in the monkey pulvinar[41]. This is species specific or dependent on the subarea of pulvinar. In mice, activation of the projection from SC to LP is enough to induce freezing[42]. And just like we found in mice, silencing the sSC in rabbits causes a sharp attenuation of visual response in LP[43]. We found, however, that silencing LP does not remove the effects of suppressing the sSC, and that in anesthetized mice, there is little change in visual response in V1 when LP is silenced. Although it is difficult to silence LP without silencing dLGN, our recordings in LP and post hoc imaging of the fluorescent muscimol diffusion convinced us that we had indeed silenced LP. Furthermore, reducing thalamic activity optogenetically using a genetic restriction to LP also showed a lack of change in the

overall response to these stimuli in V1 of anesthetized mice, although we could only achieve a relatively weak reduction of LP activity. Silencing LP by muscimol had an effect on the responses of a subset of V1 neurons (Supplementary Fig. 4D-E), but did not cause an overall reduction in V1 activity. This seems different in monkeys. In galago monkeys, V1 was silenced by silencing pulvinar[44], although deactivation of the pulvinar had only mixed effects on activity in V2 of cebus monkeys[45] and caused reduction, but not silencing, of V4 activity in rhesus monkeys[46]. Compared to primates, rodents have a relatively small LP/pulvinar complex[47] and the influence on V1 activity of LP in rodents may be less than it is in primates. Functional connectivity is also altered by anesthesia[9] and it will be interesting to investigate these interareal relationships in awake mice. It is already known that activating the connection from SC to LP induces behavior[42]. The combination of multi-areal recordings and silencing, especially those involving the triple recordings in SC, LP, and V1 (Fig. 2d), is however technically very challenging.

Optogenetically inhibiting the sSC decreased the responses in the dLGN, but not the relative size tuning profile. We believe that the change in size tuning in V1 is a direct effect of the reduced responses in the dLGN. The size tuning of primate V1 neurons differs depending on the stimulus contrast and there is more spatial summation for lower contrast stimuli[48,49]. We have confirmed earlier reports that in mouse V1, surround suppression is

also higher for higher contrasts[50,51]. The changes in dLGN and V1 responses resulting from sSC suppression were not distinguishable from a change in stimulus contrast, or a dLGN gain change. We therefore interpret the primary effect of superior colliculus on the responses in V1 as a local gain modulation. The resulting change of surround suppression in V1 can be explained by interpreting surround suppression as an effect of normalization[52]. Normalization describes many aspects of responses in the visual system, also in mouse V1[50,53]. The mechanisms underlying normalization, however, are not fully understood[52]. Non-GABAergic mechanisms will be involved in other aspects of normalization[54], but inhibition plays a role in V1 surround suppression[55]. This was shown in particular for inhibition from somatostatin-positive interneurons when mice are running[56], although the mechanisms could be different when mice are not running[57] and surround suppression is increased[50]. Like running, anesthesia can decrease the level of surround suppression[56,58,59] depending on its depth[60]. Normalization mechanisms operate under anesthesia[61], but changes in contrast sensitivity and response gain due to the level of anesthesia[58] will influence surround suppression. The level of surround suppression therefore depends on many factors during an experiment. This makes it important to use fast control of the activity in the sSC using optogenetics.

The superior colliculus has a well-established role in orienting and directing attention in response to external stimuli[15]. The basis for this may be the computation of a saliency map in the sSC[17], where stimulus features with a higher saliency have a higher neural response. We have found that higher firing rates in the sSC result in an increased gain in the shell of the dLGN at the matching retinotopic location. The increased response in dLGN for the feature location leads to an increased response in V1. Firing rates in primate V1 correlate with perceptual saliency[62–65], and these increased responses may lead to faster behavioral responses, lower detection thresholds, and more accurate processing. In the mouse, a saliency map computed in the sSC may enhance the responses to salient stimulus parts in V1. The computation of stimulus saliency in V1 itself may have become more explicit in the primary visual cortex during primate evolution[16]. This may explain why we and others observe changes in visual cortex responses after silencing SC in the mouse[11], while there is no change in attentional modulation in middle temporal/medial superior temporal (MT/MST) if SC is inactivated in the primate[28]. The lack of change in attentional modulation in MT/MST could also be an illustration of the difference between top-down attention, as used in the primate task, and bottom-up exogenous attention brought about by the stimulus itself. Regardless of this, in monkeys there is input from the SC to the dLGN[13] and thus sSC may also have a similar effect on dLGN gain and bottom-up modulation of primate V1 responses. Another possibility is that the response reduction that we found in V1 is the effect of disabling a feedforward retinotectogeniculate stream of input to V1, which provides input in parallel of the direct retinogeniculate pathway to V1. Future investigation of responses of dLGN neurons receiving tectal input would provide insight into this question.

## Methods

**Animals and regulations.** C57BL/6JOlaHsd (Harlan) and C57BL/6J (Janvier) mice, Gad2-Cre+ mice[19], Calb2-Cre+ mice[19], and Gad2-Cre+ crossed to Ai14 tdTomato reporter line[66] mice of 2–5 months of age were used for the experiments. Male and female mice were used. Mice were housed in a 12 h/12 h dark/light cycle with access to food and water ad libitum. All experiments were approved by the institutional animal care and use committee of the Royal Netherlands Academy of Arts and Sciences and complied with all relevant ethical regulations.

**Electrophysiology surgery for anesthetized recording.** Mice were anesthetized by an intraperitoneal (IP) injection of 1.2 g urethane per kg body weight, supplemented by an IP injection of 8 mg chlorprothixene per kg body weight. We injected atropine sulfate (0.1 mg per kg) and dexamethasone (4 mg per kg) subcutaneously (s.c.) to reduce mucous secretions and to prevent cortical edema, respectively. Additional doses of urethane were injected when a response to a toe-pinch was observed. Mice were head fixed by ear and bite bars. Temperature was measured with a rectal probe and maintained by a feedback-controlled heating pad set to 36.5 degrees.

**Electrophysiology surgery for awake recording.** Mice were first anesthetized with isoflurane (5% induction, 1.2–1.5% maintenance) in oxygen (0.8 L per min flow rate). Rectal temperature was maintained at 36.5 ℃. The eyes were protected from light by black stickers and from drying by Cavasan eye ointment. During their surgery, mice were administered the analgesic Metacam (1 mg per kg s.c.) to reduce pain during the recovery. Mice were head fixed and the scalp and soft tissue overlying the skull were incised to expose the skull. A metal ring (5 mm inner diameter) was attached to the skull with glue and dental cement. Small craniotomies for recording were made by dental drill. Next, the head was fixed to a stand through a handle attached to the ring. Animals recovered for 2 h before the recordings started. The animals were given water and milk in the first hour after recovery, while they were restrained. Animals were killed at the end of the recording session by an overdose of pentobarbital (100 mg per kg IP).

**Electrophysiological recording and analysis.** Laminar silicon electrodes (A1× 16-5mm-50-177-A16, 16 channels spaced 50 μm apart, Neuronexus) were used for all the extracellular recordings from sSC, V1, LP, dLGN, and PBG. For recordings in the sSC, electrodes were inserted through a craniotomy 600–800 μm lateral and 600–800 μm anterior and 1000–1800 μm down from the Lambda landmark. There was considerably less visual response below 300 μm from surface of the sSC (where the first visual response is observed). This means usually around 7–8 channels of the laminar probe were within the sSC. For recordings in V1, electrodes were inserted through a craniotomy 2900–3000 μm lateral and 300–500 μm anterior to Lambda and 0–800 μm down from the cortical surface. For recordings in dLGN, electrodes were inserted through a craniotomy 2050–2200 μm lateral and 2450–2550 μm posterior to Bregma and 2200–3000 μm down from the cortical surface. For LP recordings, electrodes were inserted through a craniotomy 1700–1800 μm lateral and 2200–2350 μm posterior to Bregma and 2100–2900 μm down from the cortical surface. PBG is a very small midbrain nucleus located at the most lateral side of the midbrain, and targeting the laminar probe only based on the stereotactic coordinates had a low success rate. Therefore, in order to increase the success rate, we developed a sensory guided method together with the stereotactic coordinates to target the PBG. Hence, for the PBG recordings, a craniotomy 1900–2100 μm lateral and 50 μm posterior to 100 μm anterior to Lambda was made and the electrode was sent down through the craniotomy, while waving a hand in front of the mouse eyes to produce a visual response (as the craniotomy is above monocular V1) till 700–800 μm down from the cortical surface. If the electrode was in the correct location in the craniotomy, starting at depth of 1900–2000 μm, sounds in frequency range of 1–3 kHz (at higher than 50 dB) produced an auditory evoked potential (the location is in a part of the external nucleus of inferior colliculus sensitive to this specific sound, based on our experience). Getting this evoked potential earlier than 1900 μm deep meant that the probe should be relocated more lateral. Getting the evoked potential only below 2000 μm meant that the probe should be relocated more medial within the craniotomy.

**Visual stimulation for electrophysiology.** Stimuli were projected by a gamma-corrected PLUS U2-X1130 DLP projector onto a backprojection screen (Macada Innovision), positioned 17.5 cm in front of the mouse. Full screen size was 60 × 42 cm. Matlab Psychophysics Toolbox 3 was used to generate the visual stimuli[67]. In order to find the RF position of the sSC, V1, and dLGN neurons, we presented a 5 min movie (5 frames per second) of small white squares (approximately 5 degrees wide) in random positions on black background (ratio of white to black area: 1:30). As the PBG RFs were large, larger white squares were presented (15 degrees) to find the RF positions of the PBG neurons. Presenting these stimuli was not sufficient to determine RFs of the LP neurons, as the LP RFs are large and probably the LP neurons were not well activated by these stimuli. Therefore, in order to determine the LP RFs, we presented drifting gratings of 0.05 cycles per degree in large square patches (15 degrees) with 25% overlap between the positions (10 times repetitions for each position). In order to measure size tuning and surround suppression index, disks of square wave gratings drifting in different directions (0–330 with steps of 30 degrees) centered at the RF positions with a fixed range of physical diameters. These diameters corresponded to 10, 25, 40, 60, 90, and 120 degrees of visual angle when the stimuli were shown directly in front of the mouse. For some penetrations, the RF positions for deeper V1 layers were different from the top layers. In these cases, the series of size tuning stimuli was run twice with different centers, and only the data from the layers with an RF close to the presented stimulus center were included. The gratings had a spatial frequency of 0.05 cycles per degree and a 95% contrast. The physical spacing of the grating was fixed across

the screen and the spatial frequency was computed at the center of the screen. The temporal frequency was 2 Hz. Stimulus duration and interstimulus time were both 1 s. Background luminance was 10 cd per m². For comparing the effects of reducing contrast and optogenetic inhibition of the sSC, we determined the size that gave, on average, the best response for all simultaneously recorded V1 units. At this size, we then measured contrast tuning by showing the gratings at 5, 20, 35, 50, 65, 80, and 95% contrast, with and without optogenetic inhibition of the sSC. From these curves, we determined the highest contrast at which the laser caused a roughly 15% response reduction (Fig. 5a). We then picked as the lower contrast, a contrast at which the response with the laser off was roughly similar to the response at high contrast with the laser on. For three penetrations, this meant a high contrast of 90% and a lower contrast of 70%, for one penetration, this was 95% versus 65%, and for one penetration this was 75% versus 65%.

**Analysis of electrophysiology data.** Laminar probe signals were amplified and filtered at 500 Hz–10 kHz and digitized at 24 kHz using a Tucker-Davis Technologies RX5 pentusa. Signals were thresholded at 3× standard deviation to isolate spikes, and spikes were sorted by custom-written Matlab (Mathworks) scripts, but single and multi-units were pooled together for this publication to increase the number of measurements, except for Supplementary Figures 6A-B which only contain single units. When we write response for the extracellular recordings, we mean the evoked visual response, averaged over the duration of the stimulus, minus the spontaneous rate. The spontaneous rate was defined as the mean rate in the last 0.5 s before stimulus onset. Minimum response for a unit to be included was 2 Hz. We quantified surround suppression with a surround suppression index (SSI), defined as $(R_{pref} - R_{large})/R_{pref}$, where $R_{pref}$ is the response to the preferred stimulus size, averaged over all directions, and $R_{large}$ is the response to the largest size, averaged over all directions. For calculation of any effects on surround suppression, we excluded the cells that were not surround suppressed (SSI = 0) in the laser-off condition, because a change in the relative response between optimal and large stimuli may not induce a change in the SSI of a cell that is not suppressed. This was primarily done to make sure that we did not miss a change in surround suppression in the dLGN when optogenetically inhibiting the sSC. For the dLGN recording shown in Fig. 3g, it meant excluding 11 of the 37 cells, because there is much less surround suppression in the dLGN. The change in dLGN SSI remains not significant when we include these cells ($p = 0.48$, Wilcoxon test, 37 units). For the V1 recording shown in Fig. 1h, we have done the same for consistency, but it meant excluding only 5 of the 165 cells. The decrease in SSI is also significant when we include these cells ($p = 6.3 \times 10^{-11}$, Wilcoxon test, 165 units). The OSI was defined as $OSI = \sqrt{(\Sigma R(\varphi) \sin(2\varphi)^2 + \Sigma R(\varphi) \cos(2\varphi)^2)} / \Sigma R(\varphi)$, where $\varphi$ is the direction of the stimulus and $R(\varphi)$ the neuron's response. This is equal to 1–circular variance. DSI was defined by $DSI = \sqrt{(\Sigma R(\varphi) \sin(\varphi)^2 + \Sigma R(\varphi) \cos(\varphi)^2)} / \Sigma R(\varphi)$.

**Optogenetics and drug delivery.** For inhibiting the sSC by optogenetics, Gad2-Cre mice were anesthetized with isoflurane (5% induction, 1.5–2.5% maintenance) and three small craniotomies were made above the sSC. Using a Drummond Nanoject volume injector at two depths of the sSC, 46 nl (in each) was injected of a solution of Cre-dependent adeno-associated virus with a ChR2 vector with Ef1a promoter (AAV9.Ef1a.DIO.hChR2(H134R)-eYFP.WPRE.hGH, $1.5 \times 10^{13}$ genome copies (GCs) per ml, University of Pennsylvania Vector Core). The scalp was resutured and the vector was allowed to express for 4–5 weeks before acute electrophysiology. A blue fiber-coupled laser (473 nm, DPSS Laser T3, Shanghai Laser & Optics Co.) was used to activate the ChR2. The fiber ended in the craniotomy over retrosplenial cortex above the sSC. Trials with laser on were intermingled with trials with laser off. In laser-on trials, the laser was on 1 s before stimulus onset until stimulus offset. In order to have a broad silencing of LP or PBG, we injected about 100 nl or 70 nl (respectively) of 2.5 mM fluorescent-conjugated muscimol (an agonist of GABA_A receptors; Life Technologies) in these brain regions by a Drummond Nanoject volume injector (with volume rate of 2.3 nl per second). The targeting of these areas followed the targeting of the recording electrodes.

For optogenetically decreasing the LP activity in Supplementary Figure 5, Calb2-Cre mice were injected by Cre-dependent AAV with an Arch vector (AAV1.Flex.CBA.Arch-GFP.WPRE.SV40, $1.1 \times 10^{13}$ GCs per ml, University of Pennsylvania Vector Core) in 2 depths in LP (about 180 nl). The scalp was resutured and the vector was allowed to express for 4–5 weeks before acute electrophysiology. A green fiber-coupled laser (532 nm, DPSS Laser T3, Shanghai Laser & Optics Co.) was used to activate the Arch. The fiber was inserted in the brain but kept 1.5 mm above the LP to cover the whole LP by laser light. Trials with the laser on were intermingled with trials in which the laser was off. In laser-on trials, the laser was on 1 s before stimulus onset until stimulus offset.

**Statistics.** Throughout the manuscript, averages are given as mean ± s.e.m. For significance testing, we first tested if the data from all groups was normally distributed using the Shapiro–Wilk test at the 5% rejection level. If all data were normally distributed, we used parametric testing, and otherwise we used nonparametric testing. All compared groups had no significantly different variances. For pairwise comparisons, we used parametric paired $t$-tests for normally distributed, and nonparametric Wilcoxon signed rank tests otherwise. For unpaired testing, we used parametric $t$-tests and nonparametric Mann–Whitney tests for two

groups, and ANOVA and Kruskal–Wallis for comparing multiple groups. For testing if a manipulation had a significantly different effect over a range of tested stimuli, we looked at the interaction in a two-way ANOVA. All tests were done two-sided. The number of mice used for each experiment was determined by previous experience.

**Generation of CAV2-ZsGreen.** pCAV2-ZsGreen was generated by subcloning ZsGreen cDNA (Clontech) into pTG5412/CAG-MCS[68], using *Fse*I and *Pme*I restriction sites. Recombinant clones were screened by ampicillin resistance and chloramphenicol sensitivity. Putative recombinant clones were verified by restriction digest with *Sce*I and *Eco*RI and DNA sequencing. pCAV2-ZsGreen was transfected into DK-SceI cells[69] using lipofectamine LTX (Invitrogen) according to the manufacturer's instructions. pCAV2-ZsGreen was linearized following transfection by 1 h of treatment with (z)-4-OH-tamoxifen (Tocris) at 0.5 M. At 72 h post transfection, viral particles were liberated by freeze thaw and serially amplified in DK-SceI cells. Amplified viral particles were purified by serial CsCl gradient centrifugation[70].

**Viral tracing.** For Fig. 3a and Supplementary Figure 7, 100 nl of a retrograde virus CAV2.CAG.ZsGreen was unilaterally injected in the right dLGN (2200 μm lateral and 2500 μm posterior to Bregma in depths of 2500 μm and 2700 μm from the cortical surface) of Gad2-tdTomato mice. For Fig. 3b, 46 nl of AAV5.CAG.tdTomato.WPRE (UNC Vector Core) and 46 nl of AAV5.CAG.GFP.WPRE (UNC Vector Core) were unilaterally injected in two different locations in the sSC (GFP: 500 μm lateral and 600 μm anterior to Lambda in depths of 1300 μm and 1600 μm from the cortical surface; tdTomato: 900 μm lateral and 200 μm anterior to Lambda in depths of 1400 μm and 1700 μm from the cortical surface) of C57BL/6J (Janvier) mice. At 3 to 4 weeks after the injection, the mice were perfused.

**Perfusion and microscopy.** After an overdose of pentobarbital (100 mg per kg IP), mice were transcardially perfused with 4% paraformaldehyde (PFA) in phosphate-buffered saline (~80 ml per mouse) and postfixed for 2 h in PFA at 4 ℃. Using a vibratome (Leica VT1000S), the perfused brains were sliced coronally with thickness of 50 μm. For quantification of the ZsGreen-positive cells co-labeled with tdTomato in Fig. 3a, the brain slices were imaged by a 20× objective (NA 0.7) on a Leica SP5 confocal microscope. Brain slices represented in other figures were imaged by a Zeiss (Axioplan2) fluorescence microscope.

**Code availability.** All codes used for generation of the stimuli and analyses of the data are available at http://github.com/heimel/InVivoTools.

## Data availability
The datasets generated during and/or analyzed during the current study are available from the corresponding author on reasonable request.

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

## Acknowledgements

This work was funded by NWO VIDI grant 864.10.010 to J.A.H. We thank Christiaan Levelt for sharing mice and equipment, Emma Ruimschotel for genotyping, Rudolf Faust for help with viruses, Ralph Hamelink for optical fiber coupling, and Pieter Roelfsema for sharing equipment. We are grateful to Z. Josh Huang and his colleagues from the NIH Neuroscience Blueprint Cre Driver Network and Hongkui Zeng of the Allen Institute for Brain Science (AIBS) for creating and making their mouse lines available and to Ed Boyden and the Massachusetts Institute of Technology for providing viral vectors.

## Author contributions

M.A. co-designed, performed, and analyzed the experiments and co-wrote the paper. L.S.Z. made the CAV2 virus. J.A.H. co-designed the experiments and co-wrote the paper.

## Additional information

**Competing interests:** The authors declare no competing interests.

