## [Peer Review File · Nature Communications]

Reviewers' comments:

Reviewer #1 (Remarks to the Author):

Beautiful work and an important step forward. The functional significance of SC outputs to dLGN (shell/Clayers) and LP has been cryptic. This paper includes many important features; verification it is excitatory, influence on V1 rfs, etc., and arrives there with rigor in tools and analysis. Impressive and characteristic of this group.

One issue that needs resolving however: the superficial SC and dLGN outer shell are known to receive On-Off and J RGC inputs (see Dhande and Huberman, Curr Opin 2014 or Seabrook et al Ann Review of Neuro 2017) and to project to superficial layers of V1 (if not exclusively, then at least preferentially). The current results should be discussed in light of the preponderance of DS inputs and responses (Piscopo et al., J Neuro, 2013) and since Clayers and tecto-recipient dLGN is also equivalent to konio dLGN in monkey, the broader context of that circuitry too. That aside, no specific issues arise. Data are clear, as is the writing. A high quality manuscript sure to make a strong contribution to the literature.

Reviewer #2 (Remarks to the Author):

This paper presents a clever dissection of the contribution of Superior Colliculus (SC) input to V1 in the mouse, using a combination of electrophysiological recording and optogenetic as well as pharmacological manipulations. The principal findings – that SC input selectively boosts responses in V1 to optimum sized stimuli, that it does so via the tectogeniculate pathway, and that the gain modulation via the LGN is itself size-independent, are novel and make a significant contribution to understanding SC function in rodents.

My only major criticism is that it isn't made clear why the whole study wasn't carried out on awake animals. The main effect (reduction in response to optimum sized stimulus when SC inhibited optogenetically) is shown for both anesthetized (Fig. 1) and awake animals (Fig. S2), and appears similar, but later the authors state that "under anesthesia there is less suppression than in awake animals" and mention the "dependence of size tuning on the state of the animal" (p.11).

The authors should also comment more explicitly on the cellular mechanism of the size tuning. They mention briefly the normalization model (Carandini & Heeger 2011) but could also discuss that their data does not support inhibition (e.g. by GABA) as an explanation for surround suppression.

Minor points:

- 1) P.7, l.20: "There is also change in the ratio..."; insert no (no change!)
- 2) P.8, l.6: "the response V1"; insert in
- 3) P.9, l.17: "the functional effect is often superior colliculus..."; this sentence is garbled
- 4) P10., l.26-27: "Optogenetically inhibiting the sSC decreased the responses in the dLGN, but not the relative size tuning profile. We believe that the this change in size tuning in V1..."; this sounds contradictory and is not very clear (esp. "that the this")
- 5) P.10, l.29: "... there is more spatial summation for lower contrast stimuli"; this was first shown by Sengpiel et al. (1997).
- 6) P.11: "response reduction V1..."; delete V1
- 7) P.18, l.10: "... the smallest size presented within a recording"; weren't the sizes tested the same for all stimuli? This is the impression I got from the Methods (p.29, l.1)

Reviewer #3 (Remarks to the Author):

In this manuscript Ahmadlou et. al demonstrate that inhibiting the activity of neurons in the mouse superior colliculus (SC) changes the responses of V1 neurons to light stimulus, specifically

by affecting the surround suppression characteristics of V1 neurons. The authors then aim to determine which pathway (dLGN, LP, or PGB) is responsible for this change. The latter two areas were silenced with muscimol while optogenetically silencing the SC and recording visual response properties of V1. This experiment showed that the LP and PGB inhibition did not change the effect of silencing the SC on V1 response properties. Direct recording of the dLGN while silencing the SC revealed a lowered response of dLGN neurons (change in gain) to visual stimulation. The change in dLGN and V1 response properties observed when silencing the SC, were mimicked by altering the contrast of the visual stimulus. The authors conclude that the mouse SC can modulate V1 response properties in the mouse via a dLGN intermediate. This work is significant because it demonstrates that the mouse SC, which gets the majority of visual input in the mouse, can contribute to V1 visual processing, a novel finding that will be of interest to many researchers across the world and will inspire future work to understand how these changes relate to behavior. However, there are a number of clarifications about the statistics and data presentation that need to be addressed to convince me that the author's conclusions are sound.

1. The authors state that the silencing the sSC by Gad2 cell activation modulates the V1 response, but silencing the LP by muscimol does not. However, these two statistical tests were not conducted in a comparable manner. First, the authors used a larger sample size ($n=160$ for sSC silencing vs. $n=42$ or $n=49$ for LP), which makes the LP silencing result more unlikely to be significant. Second, the statistics for the sSC silencing is done with the Wilcoxon signed rank test (paired) while that for the LP silencing is done with ANOVA with a different set of cells. Both of these points biases the results toward having a non-significant result for the LP silencing. The authors should use the same treatment of data for the two experiments. For example, use a randomly sub-sampled equal number of neurons from both the sSC and LP silencing experiments and use ANOVA for both of them, or conduct paired recording in the LP silencing experiment. In addition, the supplementary data provided in Figure S5 does not help much because the reduction of the LP cell firing rate is small (so the net effect on V1 should be also small) and, again, the statistics are low.

2. The PBG silencing experiment has the same issue of the inconsistent sample sizes. In addition, in figure 4I, there seems to be a systematic firing rate change by the PBG silencing. With the PBG modulation, the V1 responds more to small stimuli and less to large stimuli. Is the significance of this effect evaluated? Adding a simple parametric model to describe stimulus size dependence of the FR will be interesting. This is important because the PBG potentially has a different type of effect on V1 either through its direct connection to the dLGN or reciprocal connection to the sSC.

3. There is no indication of the cortical layers that the recorded V1 cells belong to. These are important because of different areas of the brain send projections to different layers. For example the dLGN shell is reported to project to more superficial layers (Cruz-Martin et al. 2014), thus there may be significant differences between neurons not seen in the present data. Please indicate where the cells were recorded. Moreover, it will increase the significance of the data if the modulation of the V1 cells is layer specific.

Minor concern

The authors did not evaluate the statistical significance of the orientation/direction selectivity (Figure S4). If the non-OS/DS cells were included in the statistics, it biases the results toward being non-significant. The authors should first pick up only statistically significant OS/DS cells and then determine if the tuning properties change or not. This may reveal something interesting, especially given the Cruz-Martin result.

Typos

Page 2, Line 18: it is has -> it has

Page 9, Line 35: the monkey pulvinar -> of the monkey pulvinar
Page 10, Line 27: the this -> this

Point-by-point reply

We thank all three reviewers very much for their critical reading of our manuscript, their positive comments and their helpful suggestions. We have done new recordings to better compare the effects of LP silencing and sSC suppression on responses in V1. We have made additional analyses and extended the discussion as requested by the reviewers.

Reviewer #1 (Remarks to the Author):

Beautiful work and an important step forward. The functional significance of SC outputs to dLGN (shell/Clayers) and LP has been cryptic. This paper includes many important features; verification it is excitatory, influence on V1 rfs, etc., and arrives there with rigor in tools and analysis. Impressive and characteristic of this group.

One issue that needs resolving however: the superficial SC and dLGN outer shell are known to receive On-Off and J RGC inputs (see Dhande and Huberman, Curr Opin 2014 or Seabrook et al Ann Review of Neuro 2017) and to project to superficial layers of V1 (if not exclusively, then at least preferentially). The current results should be discussed in light of the preponderance of DS inputs and responses (Piscopo et al., J Neuro, 2013) and since Clayers and tecto-recipient dLGN is also equivalent to konio dLGN in monkey, the broader context of that circuitry too.

This is indeed an interesting aspect that we had ignored in the manuscript. In response to this, we have moved the supplemental figure on the lack of change in orientation and direction selectivity in V1 to after the results that show that the collicular modulation of V1 is mediated by the dLGN. There, we have added an explanation with the context provided by the reviewer on why it is interesting to look at these aspects. We have extended the analysis of the orientation and direction selectivity changes in V1, by including only a subset of the sensitive cells (following a suggestion of reviewer 3). As the reviewer suggest, the dLGN projects preferentially to the superficial layers of V1. For this reason, one would perhaps expect the superficial layers of V1 to show the most change in response to the optogenetic suppression of sSC. We have added a panel (Fig. 11) with a categorization of the changes by depth, showing that, perhaps surprisingly, this is not the case.

In addition, we have included a discussion of these aspects in the Discussion, which reads *“The shell of the dLGN also receives direct input from different types of direction-selective retinal ganglion cells^{25,34-36} and contains a higher proportion of orientation- or direction-selective relay cells than the dLGN core²⁶. The shell also has a high number of morphologically W-like cells³⁷ that receive retinal and tectal input¹². Rodent W-like cells are likely to be homologous to carnivore W and primate koniocellular cells³⁸. The preponderance of W-like cells and its connectivity pattern suggests a homology of the mouse shell to the carnivore C-layers and primate koniocellular layers²⁵. The relay neurons of the shell preferentially target the superficial layers of V1 and transmit this direction and orientation selectivity^{27,28}. Surprisingly, however, we found neither changes in orientation or direction selectivity in V1, nor a larger effect in superficial V1.”*

That aside, no specific issues arise. Data are clear, as is the writing. A high quality manuscript sure to make a strong contribution to the literature.

Reviewer #2 (Remarks to the Author):

This paper presents a clever dissection of the contribution of Superior Colliculus (SC) input to V1 in the mouse, using a combination of electrophysiological recording and optogenetic as well as pharmacological manipulations. The principal findings – that SC input selectively boosts responses in V1 to optimum sized stimuli, that it does so via the tectogeniculate pathway, and that the gain modulation via the LGN is itself size-independent, are novel and make a significant contribution to understanding SC function in rodents.

My only major criticism is that it isn't made clear why the whole study wasn't carried out on awake animals. The main effect (reduction in response to optimum sized stimulus when SC inhibited optogenetically) is shown for both anesthetized (Fig. 1) and awake animals (Fig. S2), and appears similar, but later the authors state that "under anesthesia there is less suppression than in awake animals" and mention the "dependence of size tuning on the state of the animal" (p.11).

We had briefly mentioned the primary reason in the original discussion section: "The combination of multi-areal recordings and silencing is however technically very challenging in the awake animal because of the duration and invasiveness of the procedures." The duration of the procedure is the key issue here. Our local ethical committee allows only head-fixation sessions up to about 3 hours. For very specific questions or reasons, they can make exceptions to this, but their expectation is that longer head-fixed sessions cause the mouse much discomfort. Many of the experiments done in this manuscript, however, require much more time than this. In particular, the experiment illustrated in Figure 2D, in which we evaluate the effect of collicular silencing before and after silencing LP takes at least eight hours. The reason for this is that we need to insert electrodes into three retinotopically matching locations, insert a fiber above the SC, record, retract the cortical electrode, inject muscimol in LP, reinsert the cortical electrode and record again. Given the small size of the mouse head, the procedure is quite challenging. Chronically implanted electrodes would speed up the recording procedure, but we are currently not capable of getting good signal out of three chronically implanted electrodes. Therefore, in our lab we had no alternative than to do these experiments to investigate the circuitry underlying the collicular modulation of V1 under anesthesia.

We think that adding this whole paragraph to the Discussion would be too long, but we have extended it to: "*The combination of multi-areal recordings and silencing, especially those involving the triple recordings in SC, LP and V1 (Fig. 2D), is however technically very challenging in the awake animal because it requires head-fixation for many consecutive hours, unless the technical challenge of chronically recording from three retinotopically matched positions is solved.*"

In addition, we have rewritten and extended the discussion on the level of surround suppression. It now reads: "*Like running, anesthesia can also decrease the level of surround suppression^{59,61,62} depending on the depth of the anesthesia⁶³. Normalization mechanisms still operate under anesthesia and much of the original evidence for the normalization model was obtained in anesthetized animals⁶⁴. The state of the animal, however, also modulates contrast sensitivity and response gain⁶¹. Modulation of these quantities will influence surround suppression, as we and others have shown how contrast influences surround suppression. The absolute level of surround suppression is therefore likely to depend on many factors during an experiment. This makes it particularly important to use the possibility of fast switching of the activity in the sSC using optogenetics.*"

The authors should also comment more explicitly on the cellular mechanism of the size tuning. They mention briefly the normalization model (Carandini & Heeger 2011) but could also discuss that their data does not support inhibition (e.g. by GABA) as an explanation for surround suppression.

We have added to following to the discussion: *“The resulting change of surround suppression is an effect of local mechanisms operating in V1 and can be explained by interpreting surround suppression as an effect of normalization⁵⁵. The phenomenological normalization model accurately fits many aspects of visual response in the monkey and carnivore brain, and was shown also to fit mouse V1 response properties^{53,56}. The mechanisms underlying normalization are not fully understood⁵⁵. Non-GABAergic mechanisms will be involved in other aspects of normalization⁵⁷, but inhibition is thought to play a role in V1 surround suppression⁵⁸. This was shown in particular for inhibition from somatostatin-positive interneurons when mice are running⁵⁹, although the mechanisms could be different when mice are not running⁶⁰ and surround suppression is increased⁵³. ”*

We hope that the reviewer can agree with this interpretation. This does not directly say that our data does not support inhibition as an explanation for surround suppression, because we think that we do not have data to support other mechanisms.

Minor points:

- 1) P.7, l.20: *“There is also change in the ratio...”*; insert *no (no change!)*

Corrected.

- 2) P.8, l.6: *“the response V1”*; insert *in*

Inserted.

- 3) P.9, l.17: *“the functional effect is often superior colliculus...”*; this sentence is garbled

Corrected. ‘often’ should have been ‘of the’

- 4) P10., l.26-27: *“Optogenetically inhibiting the sSC decreased the responses in the dLGN, but not the relative size tuning profile. We believe that the this change in size tuning in V1...”*; this sounds contradictory and is not very clear (esp. *“that the this”*)

Removed spurious ‘this’

- 5) P.10, l.29: *“... there is more spatial summation for lower contrast stimuli”*; this was first shown by Sengpiel et al. (1997).

The reviewer is right. This was an oversight. We have added the reference.

- 6) P.11: *“response reduction V1...”*; delete V1

Deleted.

- 7) P.18, l.10: *“... the smallest size presented within a recording”*; weren’t the sizes tested the same for all stimuli? This is the impression I got from the Methods (p.29, l.1)

We indeed showed physically exactly the same size range of stimuli for each recording, but because the stimuli were for each recording centered at a slightly different place in the visual field, the area of visual angle the stimuli covered was not the same for each recording. We had tried to describe in the Methods by “centered at the RF positions with fixed diameters

corresponding to 10, 25, 40, 60, 90 and 120 degrees of visual angle when shown directly in front of the mouse.” This was too cryptic. We have changed this to: “*centered at the RF positions with a fixed range of physical diameters. These diameters corresponded to 10, 25, 40, 60, 90 and 120 degrees of visual angle when the stimuli were shown directly in front of the mouse.*”

We hope this is more clear.

Reviewer #3 (Remarks to the Author):

In this manuscript Ahmadlou et. al demonstrate that inhibiting the activity of neurons in the mouse superior colliculus (SC) changes the responses of V1 neurons to light stimulus, specifically by affecting the surround suppression characteristics of V1 neurons. The authors then aim to determine which pathway (dLGN, LP, or PGB) is responsible for this change. The latter two areas were silenced with muscimol while optogenetically silencing the SC and recording visual response properties of V1. This experiments showed that the LP and PGB inhibition did not change the affect of silencing the SC on V1 response properties. Direct recording of the dLGN while silencing the SC revealed a lowered response of dLGN neurons (change in gain) to visual stimulation. The change in dLGN and V1 response properties observed when silencing the SC, were mimicked by altering the contrast of the visual stimulus. The authors conclude that the mouse SC can modulate V1 response properties in the mouse via a dLGN intermediate. This work is significant because it demonstrates that the mouse SC, which gets the majority of visual input in the mouse, can contribute to V1 visual processing, a novel finding that will be of interest to many researchers across the world and will inspire future work to understand how these changes relate to behavior. However, there are a number of clarifications about the statistics and data presentation that need to be addressed to convince me that the author’s conclusions are sound.

1. The authors state that the silencing the sSC by Gad2 cell activation modulates the V1 response, but silencing the LP by muscimol does not. However, these two statistical tests were not conducted in a comparable manner. First, the authors used a larger sample size (n=160 for sSC silencing vs. n=42 or n=49 for LP), which makes the LP silencing result more unlikely to be significant. Second, the statistics for the sSC silencing is done with the Wilcoxon signed rank test (paired) while that for the LP silencing is done with ANOVA with a different set of cells. Both of these points biases the results toward having a non-significant result for the LP silencing. The authors should use the same treatment of data for the two experiments. For example, use a randomly sub-sampled equal number of neurons from both the sSC and LP silencing experiments and use ANOVA for both of them, or conduct paired recording in the LP silencing experiment.

The most important point of the LP silencing experiment is that the modulation of V1 by optogenetic inhibition from sSC is still there, even if LP is silent (Fig. 2J). Still the reviewer is right that the evidence that we gave for an effect of sSC suppression was much stronger, than the evidence we gave for the absence of an effect of LP silencing on responses in the visual cortex. As understood by the reviewer, we cannot pair the original V1 data to before and after silencing of LP, because we could not physically fit the holders for the SC laser fiber, the SC, LP and V1 recording electrodes, and the injection pipet above the mouse skull all at the same time. To address the comment, we therefore followed the reviewer’s suggestion and performed a new set of experiments with 4 anesthetized mice, in which we did not optogenetically inhibit the superior colliculus, but measured the effect of a muscimol injection alone in LP on V1 responses. We have made a new multipanel **Supplementary Figure 4** showing this new data. Panel C shows that like the previous data there is almost no effect on the V1 population size tuning curve of the responses of all units, normalized to their firing rate

before silencing. Overall, the responses in V1 were not changed by LP silencing (optimal size, before muscimol: 57.2 ± 6.5 Hz vs after muscimol: 51.3 ± 5.3 Hz, mean \pm s.e.m., $p = 0.11$; large size, before muscimol: 48.0 ± 5.7 Hz vs after muscimol: 43.7 ± 4.8 Hz, mean \pm s.e.m., $p = 0.24$, Wilcoxon signed rank test; 4 mice, 55 units; effect of muscimol: $p = 0.55$, interaction of muscimol and size: $p = 0.95$, two-way ANOVA, **Supplementary Fig. 4A-F**). Still, **Supplementary Figure 4D-E** also show that there are individual units that are modulated by LP silencing, even in the anesthetized mouse. We have added a note of this in the Discussion.

The sample size of the new data is lower than that for the sSC silencing data. We have followed the reviewer's other suggestion, and computed the p-value for 1000 randomly sub-sampled sets of 55 units (the number of units in the new data set) for the sSC experiments (**Supplementary Fig. 4G**). All of these subsets gave a p-value smaller than 0.004. Therefore, together with the previous data from **Figure 2**, we believe that there is sufficient evidence that the effect on V1 size tuning that we found by optogenetic inhibition of sSC is not via to the tectopulvinar pathway.

Finally, we have now consistently performed two-way ANOVAs on all the effects of manipulations on size tuning and looked at the interaction terms. The p-values are included in the manuscript, if they were not present before, and were in agreement with previous statements.

In addition, the supplementary data provided in Figure S5 does not help much because the reduction of the LP cell firing rate is small (so the net effect on V1 should be also small) and, again, the statistics are low.

We agree with the reviewer that the importance of **Supplementary Figure 5** is limited. The main weakness was that the reduction of LP cell firing rate was relatively low. We were reducing the firing rate with Arch. In LP, we could not get a larger reduction, without using damaging light levels. We have been exploring other silencing opsins, but we have not managed to get stronger optogenetic silencing of LP. We could not conclude if this was an effect of the number and type of cells expressing Cre under the *Calretinin*-promoter, an effect of the used opsin, or our relative inability to get the laser light to LP. For this reason, we stopped these experiments and therefore the numbers were quite low. We still believe that including the figure in the supplementary material, as we have done, is more informative than to leave it out, but would agree to do so if asked by the reviewers or editor. To warn the reader of the somewhat limited value, we now state explicitly in the Results that the reduction was only 22% and now mention this point also in the Discussion. With addition of the extra data and analysis discussed above, we feel that overall we now show even more convincingly that this particular effect of superior colliculus on V1 responses does not involve LP.

2. The PBG silencing experiment has the same issue of the inconsistent sample sizes. In addition, in figure 4I, there seems to be a systematic firing rate change by the PBG silencing. With the PBG modulation, the V1 responds more to small stimuli and less to large stimuli. Is the significance of this effect evaluated? Adding a simple parametric model to describe stimulus size dependence of the FR will be interesting. This is important because the PBG potentially has a different type of effect on V1 either through its direct connection to the dLGN or reciprocal connection to the sSC.

To address this point, we have added a new supplementary multipanel figure 7 with more details of the effects of PBG silencing on V1 responses. We have also computed and added the statistics for the changes in V1 responses to the optimal, small and large size stimuli to the Results section. They show that there are no significant effects of silencing PBG on V1 in our data set (optimal size, before muscimol: 45.3 ± 6.1 Hz vs after muscimol: 42.2 ± 5.1 Hz,

mean \pm SEM; $p = 0.22$, Wilcoxon signed rank test; 4 mice, 43 units; small size, before muscimol: 6.8 ± 1.6 Hz vs after muscimol: 6.2 ± 1.4 Hz, $p = 0.10$; large size, before muscimol: 35.2 ± 4.8 Hz vs after muscimol: 31.9 ± 3.7 Hz, $p = 0.12$). We agree with the reviewer that Figure 4I suggest that there could be a differential effect of PBG silencing with size, but because the effects of none of the sizes are significant, we think that we cannot make much of this effect in the context of this manuscript. The p -values of 0.10 of 0.12 suggest that there may be an effect, but the number of units is already 43, so a large number of extra experiments would be needed to make the effect of the PBG on V1 by itself significant. The effect of optogenetic inhibition of sSC on V1 is much more robust. When we resample the sSC silencing data a 1000 times by randomly selecting 43 cells from the original dataset, and recompute the p -value for the Wilcoxon signed rank test on the change in V1 optimal size response we get all p -values below 0.01 (**Supplementary Fig. 7D**). This is added to the Results section.

3. There is no indication of the cortical layers that the recorded V1 cells belong to. These are important because of different areas of the brain send projections to different layers. For example the dLGN shell is reported to project to more superficial layers (Cruz-Martin et al. 2014), thus there may be significant differences between neurons not seen in the present data. Please indicate where the cells were recorded. Moreover, it will increase the significance of the data if the modulation of the V1 cells is layer specific.

We had previously looked at the modulation across depths in V1. We found no significant differences, and had therefore decided not to include it. We agree with the reviewer, however, that this is an interesting analysis, and realize that this is a somewhat surprising result, given the preferential connectivity of the sSC-recipient shell region of the dLGN to L1. We have now added a new panel (**Fig. 1I**) with the effects in V1 in at three depths and describe the results in the Results section and discuss why this is surprising in the discussion (see also our reply to Reviewer 1).

Minor concern

The authors did not evaluate the statistical significance of the orientation/direction selectivity (Figure S4). If the non-OS/DS cells were included in the statistics, it biases the results toward being non-significant. The authors should first pick up only statistically significant OS/DS cells and then determine if the tuning properties change or not. This may reveal something interesting, especially given the Cruz-Martin result.

In addition to looking at only the mean DSI and OSI of all cells in V1, we have now also looked the changes induced by optogenetic suppression of the sSC in the subsets of V1 cells that are direction-sensitive ($DSI > 0.1$) or orientation-sensitive ($OSI > 0.2$). We again found no significant changes. We have included this in the Results section and also discuss this result in the Discussion in the context of the Cruz-Martin result and the direction/orientation preference in the sSC-recipient dLGN shell. This also relates to the question of Reviewer 1 about this circuitry.

Typos

Page 2, Line 18: it is has -> it has

Corrected.

Page 9, Line 35: the monkey pulvinar -> of the monkey pulvinar

Corrected.

Page 10, Line 27: the this -> this

Corrected.

REVIEWERS' COMMENTS:

Reviewer #1 (Remarks to the Author):

I had only one critique and the authors revised the text and data organization accordingly.

As such, I maintain that this is a very interesting paper and one that will no doubt influence the visual neuroscience and sensory neuroscience field in positive ways; the influence of SC on V1 is novel and here shown convincingly to depend on dLGN circuits. The work is rigorous and the presentation is clear.

I have no further critique. A beautiful study.

Reviewer #2 (Remarks to the Author):

All my comments have been addressed fully.

Reviewer #3 (Remarks to the Author):

The reviewers have sufficiently addressed my comments. This will be an important paper for those interested in understanding how the visual system works.